# Albumin–Globulin Score Combined with Skeletal Muscle Index as a Novel Prognostic Marker for Hepatocellular Carcinoma Patients Undergoing Liver Transplantation

**DOI:** 10.3390/jcm12062237

**Published:** 2023-03-14

**Authors:** Yang Huang, Ning Wang, Liangliang Xu, Youwei Wu, Hui Li, Li Jiang, Mingqing Xu

**Affiliations:** 1Department of Liver Surgery, Liver Transplantation Center, West China Hospital of Sichuan University, Chengdu 610041, China; 2Department of Hepatobiliary Pancreatic Tumor Center, Chongqing University Cancer Hospital, Chongqing 400030, China

**Keywords:** hepatocellular carcinoma, deceased donor liver transplantation, survival outcome, prognostic index

## Abstract

Background: Sarcopenia was recently identified as a poor prognostic factor in patients with malignant tumors. The present study investigated the effect of the preoperative albumin–globulin score (AGS), skeletal muscle index (SMI), and combination of AGS and SMI (CAS) on short- and long-term survival outcomes following deceased donor liver transplantation (DDLT) for hepatocellular carcinoma (HCC) and aimed to identify prognostic factors. Methods: A total of 221 consecutive patients who underwent DDLT for HCC were enrolled in this retrospective study between January 2015 and December 2019. The skeletal muscle cross-sectional area was measured by CT (computed tomography). Clinical cutoffs of albumin (ALB), globulin (GLB), and sarcopenia were defined by receiver operating curve (ROC). The effects of the AGS, SMI, and CAS grade on the preoperative characteristics and long-term outcomes of the included patients were analyzed. Results: Patients who had low AGS and high SMI were associated with better overall survival (OS) and recurrence-free survival (RFS), shorter intensive care unit (ICU) stay, and fewer postoperative complications (grade ≥ 3, Clavien–Dindo classification). Stratified by CAS grade, 46 (20.8%) patients in grade 1 were associated with the best postoperative prognosis, whereas 79 (35.7%) patients in grade 3 were linked to the worst OS and RFS. The CAS grade showed promising accuracy in predicting the OS and RFS of HCC patients [areas under the curve (AUCs) were 0.710 and 0.700, respectively]. Male recipient, Child–Pugh C, model for end-stage liver disease (MELD) score > 20, and elevated CAS grade were identified as independent risk factors for OS and RFS of HCC patients after DDLT. Conclusion: CAS grade, a novel prognostic index combining preoperative AGS and SMI, was closely related to postoperative short-term and long-term outcomes for HCC patients who underwent DDLT. Graft allocation and clinical decision making may be referred to CAS grade evaluation.

## 1. Introduction

HCC is the third leading cause of cancer death worldwide and primarily develops in patients with cirrhosis, especially in eastern and southeastern Asia [1]. At present, prognosis in HCC patients is still a complex challenge, as the majority of them may die due to tumor recurrence or progression. Moreover, liver failure or complications of cirrhosis will seriously threaten their lives [2,3]. Comparing treatment modalities for HCC, liver transplantation (LT) is the best option because it can not only simultaneously remove the tumors and underlying cirrhosis but also eliminate complications, such as liver failure and portal hypertension [4]. Due to strict living donating criteria and complex ethical issues, most HCC patients choose deceased donor livers in China. However, with an increase in the gap between graft supply and demand, to maximize utilization of the available donor pool, comprehensive risk stratification and optimal donor–recipient matching will benefit more from limited resources [5].

Chronic infections with hepatitis B virus and/or hepatitis C virus are the strongest risk factors for HCC. Especially in China, approximately 80% of HCC cases are associated with chronic hepatitis [6]. Increasing evidence has shown that chronic inflammation plays crucial roles in tumor progression, metastasis, and recurrence by altering the tumor microenvironment and destroying immunologic function [7,8,9]. Recently, an increasing number of inflammation-based models have been used to evaluate the prognosis of patients with HCC, such as the neutrophil-to-lymphocyte ratio (NLR), lymphocyte-to-monocyte ratio (LMR), and platelet-to-lymphocyte ratio (PLR) [9,10,11]. In addition, ALB and GLB, the two major constituents of serum proteins, are considered to play pivotal roles in the inflammatory process. ALB is not only used to monitor nutritional status but also correlated with the systemic inflammatory response by suppressing the activation of cytokines [12,13]. Conversely, GLB participates in immunity and inflammation through the regulation of inflammatory cytokines [14]. Previous studies have shown that the albumin-to-globulin ratio (AGR) is an independent prognostic factor in digestive system cancers, such as colorectal cancer, gastric cancer, and cholangiocarcinoma [15,16]. Similarly, AGS is a novel predictor that reflects the cumulative effect of both ALB and GLB on esophageal squamous cell carcinoma and non-small-cell lung cancer [17,18]. However, no report has clarified the relationship between these indicators and the outcome in patients with HCC after DDLT.

The MELD score is the most frequently used method to prioritize patients with end-stage liver disease for liver transplantation, and it can also predict prognosis [19]. Despite its strong predictive value, the MELD score underestimates disease severity in approximately 15–20% of patients with cirrhosis [20,21]. To compensate for the imperfection of the MELD score, the MELDNa and five-variable MELD score were proposed on the basis of the original MELD score [22,23]. Nevertheless, we still ignored the patient’s physical and nutritional status in concurrent cirrhosis and HCC. Indeed, sarcopenia, characterized by progressive and generalized loss of skeletal muscle mass and strength with increasing age [24], is a common but underappreciated complication of cirrhosis and cancer [25]. Sarcopenia has recently been found to predict waiting list mortality and mortality following liver transplantation [26,27,28] and is associated with posttransplant severe infections or sepsis [29], longer ICU stays, and postoperative hospital stays [30].

Therefore, our study aimed to assess the effects of AGS and SMI on the prognosis of HCC patients after DDLT. Meanwhile, CAS, a novel index from the combination of AGS and SMI, was characterized to evaluate the nutritional and inflammatory status of HCC patients and was analyzed to investigate the influence on long-term outcome in HCC patients who underwent DDLT.

## 2. Methods

### 2.1. Study Population

This study included 221 adult patients who received DDLT from January 2015 to December 2019 at the Liver Surgery and Liver Transplantation Center, West China Hospital (no prisoner’s organs were used for transplantation after 2015 in our center). All subjects over 18 years were pathologically diagnosed with HCC and conformed to the University of California at San Francisco (UCSF) criteria (a solitary lesion no more than 6.5 cm or multiple lesions no more than 3 in number, none of which were larger than 4.5 cm and total tumor size no more than 8 cm in the absence of macrovascular invasion and metastasis). Patients who had liver transplantation for acute liver failure, reduced-sized liver transplantation, or combined multivisceral transplantation were excluded. Patients with allograft nonfunction within hours after revascularization with no discernible cause and leading to retransplantation or death were excluded. Patients who had no complete CT images or medical records were excluded. This study was approved by the ethics committee of the West China Hospital, in accordance with the guidelines of the 1975 Declaration of Helsinki. Informed consent was obtained from patients.

### 2.2. Preoperative Evaluation

The demographic evaluation of recipients included age, sex, body mass index (BMI) [weight (kg)/height squared (m^2^)], and etiology of cirrhosis. Since the US instituted the MELD system in 2002 and soon thereafter, MELD-based liver allocation has been adopted throughout the world. To date, the MELD score is the basis of liver allocation policy in our center. All patients had a laboratory examination including blood tests, liver biochemistries, coagulation function, serum creatinine, serum albumin, and tumor markers, and these examinations were performed every month to update the MELD score and Child–Pugh score. Head, chest, and abdomen CT scans were performed 1 week before LT to assess the tumor characteristics, including the tumor size, tumor number, presence of macrovascular invasion, and distant metastasis. Concerning the protocol of DDLT for HCC patients, two principles were considered: (1) UCSF criteria were adopted for patient selection; (2) if the tumor burden met the Milan criteria (single nodule ≤ 5 cm or 2–3 nodules, each ≤3 cm in diameter without vascular invasion or extrahepatic metastases), the patient could enjoy the MELD score adding policy up to 22 points. The functional status of recipients can be measured by the Karnofsky performance status (KPS), a simple, 11-class scale expressed as a percentage of physical function ranging from 100% (normal, no complaints, no evidence of disease) to 0% (dead) [31], combined with characteristic complications of cirrhosis, ascites, and hepatic encephalopathy. The KPS was classified into three categories according to the patient’s self-care ability. KPS A (scoring 80–100%) could carry out normal activity and work, KPS B (scoring 50–70%) could not work but could live at home and care for personal needs, and KPS C (scoring 0–40%) could not provide self-care. Quality assessment of donor allografts included donor age, sex, steatosis, serum sodium concentration, cold ischemic time (CIT), and warm ischemic time (WIT), which were used to calculate the donor risk index (DRI), a summary metric to quantify liver allograft quality [32].

### 2.3. Diagnostic Criteria and Definitions

The preoperative diagnosis of HCC was based on the criteria defined by the American Association for the Study of Liver Diseases. The diagnosis of HCC was considered reliable when the tumor characteristics were concordant with two imaging techniques, while tumor biopsy was confined to doubtful cases. All surgical complications observed during the first 90 days after DDLT were recorded according to the Clavien–Dindo classification [33] and quantified using the comprehensive complication index [34]. Postoperative infections were diagnosed by positive results from the sampling site. OS was defined as the interval between the date of DDLT and the date of death or the last follow-up until December 2021. RFS was defined as the interval between the date of DDLT and the date of recurrence in transplanted liver or extrahepatic organs when medical tests confirmed.

### 2.4. Nutritional and Inflammation Assessment

ALB and GLB are two major components of total proteins in human serum. They are routinely measured in biochemical examination. The AGR was calculated by dividing the ALB level by the GLB level. The optimal cutoff values for ALB, GLB, and AGR were identified using ROC curve analyses. On the basis of a previous study, we defined AGS as follows: patients with both hypoalbuminemia (≤ALB cutoff value) and an elevated GLB level (>GLB cutoff value) were assigned an AGS of 2, whereas those with only one of the two abnormalities were assigned an AGS of 1, and those with normal values for both parameters were assigned an AGS of 0 [17]. An AGS of 1 or 2 was defined as high AGS, while 0 was defined as low AGS. All patients were routinely examined by CT prior to liver transplantation to assess the tumor staging and anatomy of hepatic vessels and biliary ducts. The area of skeletal mass was determined by cross-sectional CT images at the level of the third lumbar vertebra (L3) using Mimics (version 21.0, Materialise NV, Leuven, Belgium). Muscles in the L3 region include the psoas, erector spinae, quadratus lumborum, transversus abdominis, external and internal obliques, and rectus abdominis (Figure 1). SMI was calculated as follows: cross-sectional area of skeletal muscle (cm^2^)/height squared (m^2^). It was divided into two groups of low SMI (Figure 1a) and high SMI (Figure 1b) according to the cutoff value of male and female. Then, the CAS grade was defined as follows: patients with both low AGS (0) and high SMI were assigned a CAS of 1, those with both high AGS (1/2) and low SMI were assigned a CAS of 3, and the others were assigned a CAS of 2.

### 2.5. Follow-Up

Following their discharge, patients visited our outpatient clinic every week for the first 3 months, then every two weeks for 3 to 6 months, and thereafter once a month regularly. The content of rechecking included blood tests, liver and renal biochemistries, tumor markers (alpha-fetoprotein and abnormal prothrombin), blood concentration of tacrolimus (FK506), and transplantation ultrasound. CT and magnetic resonance imaging (MRI), if necessary, were performed. Once tumor recurrence, liver function abnormalities, or other emergencies occurred, patients were readmitted to the hospital for subsequent therapies.

### 2.6. Statistical Analysis

All data were collected retrospectively from the institutional electronic database and clinical correspondence. The statistical analyses were performed using SPSS (version 22.0, Chicago, IL, USA), MedCalc (version 15.2.2.0, Ostend, Belgien), and GraphPad Prism (version 8.0, San Diego, CA, USA) software. Continuous variables are presented as the mean ± standard error or median (range), and categorical variables are presented as percentages. Student’s *t*-test or the Mann–Whitney U test was used to determine the difference in continuous variables between groups, and the chi-squared test or Fisher’s exact test was used for categorical variables, as appropriate. Among CAS groups, we performed the Kruskal–Wallis H test and chi-squared test for continuous variables and categorical variables, followed by the SNK-q test and Bonferroni multiple comparisons test, respectively. The 5 year OS was chosen as the primary endpoint for the survival analyses, and the 5 year RFS was used as the secondary endpoint. ROC curves were applied to determine the optimal cutoff value, as the Youden index attained the maximum value with 2 year survival as the end point. The AUCs were compared between AGS, SMI, and CAS. Survival curves were generated by the Kaplan–Meier method and compared with the log-rank test. Univariate Cox proportional hazard regression was used to identify potentially related factors. Hazard ratios (HR) and 95% confidence intervals (CI) were estimated. The multivariate analysis included all values with *p* < 0.1 in the univariable analyses. A two-tailed *p*-value less than 0.05 was considered statistically significant.

## 3. Results

### 3.1. Patient Baseline Characteristics

As shown in Table 1, a total of 221 eligible patients were enrolled in the study consecutively and were divided into two groups according to sex. We summarized the demographic characteristics of recipients and donors, laboratory parameters, intraoperative parameters, histological and gross features of tumors, and prognostic outcomes. A total of 187 patients were males (84.6%), the median age was 50 years (range 18–69 years), and the median BMI was 22.7 kg/m^2^ (range 13.9–33.6 years). Their general status was estimated by KPS on admission, and the median value was 80% (10–100%). Thirty-four patients were females (15.4%), the median age was 50 years (range 21–69 years), and the median BMI was 22.4 kg/m^2^ (range 15.2–30.9 years). The median KPS was 70% (range 10–90%).

The etiologies of liver disease were hepatitis B (86.6% in males, 76.5% in females), hepatitis C (4.8% in males, 11.8% in females), alcohol (5.9% in males, 2.9% in females), and nonalcoholic steatohepatitis (1.1% in males, 5.9% in females). Among males, 30 patients (16.0%) and 51 patients (27.3%) had concomitant encephalopathy and ascites, respectively. Similarly, five patients (14.7%) and seven patients (20.6%) had concomitant encephalopathy and ascites, respectively, in females.

### 3.2. Clinical Characteristics Related to ALB, GLB and AGR

Figure 2 shows the distribution of preoperative ALB, GLB, and AGR levels for patients divided by survival status. A high ALB level was significantly correlated with a benefitted survival outcome (*p* < 0.001, Figure 2a). The GLB level had no significant correlation with survival outcome; nevertheless (*p* > 0.05, Figure 2b), elevated AGR had a better survival outcome (*p* < 0.05, Figure 2c). The optimal cutoff values for ALB, GLB, and AGR were identified to be 39.8 g/L, 28.6 g/L, and 1.47, respectively. Survival curves showed that patients with an AGR >1.47 (*n* = 87, 39.4%) were associated with greater OS (1, 3, and 5 year OS: 97.7%, 88.5%, and 76.8% vs. 94%, 79.8%, and 64.6%, *p* = 0.034, Figure 3a) and RFS (1, 3, and 5- year RFS: 92.7%, 83.8% and 80.8% vs. 84.6%, 71.5% and 64.1%, *p* = 0.035, Figure 3d) than patients with AGR ≤ 1.47.

### 3.3. Outcome Analyses according to AGS

As shown in Table 2, we defined AGS 0 as low AGS and AGS (1/2) as high AGS; 60 patients (27.1%) were classified as low AGS, and 161 patients (72.1%) were classified as high AGS. The low AGS group had a higher KPS score (*p* = 0.039) and lower Child–Pugh score (*p* = 0.002) and MELD score (*p* = 0.037) than the high AGS group. Moreover, a decreased rate of encephalopathy, ascites, and serum AFP ≥ 400 ng/mL were observed in patients with low AGS (*p* = 0.023; *p* = 0.048; *p* = 0.011, respectively). The other characteristics, such as recipients’ demographic characteristics, tumor number, total tumor size, differentiation of HCC, and status of microvascular invasion, were comparable between the low and high AGS groups.

In addition, the low AGS patients had a lower incidence rate of postoperative infection (*p* = 0.011) and shorter duration of ICU stay (*p* = 0.039); considering all complication profiles, the low AGS group’s 90 day comprehensive complication index (CCI) was significantly lower than the high AGS group (*p* < 0.001). In addition, patients in the low AGS group had significantly longer OS (1, 3, and 5 year OS: 96.6%, 89.8%, and 80.2% vs. 95.7%, 81.3%, and 65.4%, *p* = 0.024, Figure 3b) and RFS (1, 3, and 5 year RFS: 96.6%, 88.1%, and 84.2% vs. 84.4%, 71.7%, and 65.9%, *p* = 0.011, Figure 3e).

### 3.4. Outcome Analyses according to SMI

According to the optimal cutoff values for SMI in males (43.1 cm^2^/m^2^) and females (32.9 cm^2^/m^2^). A total of 128 patients (57.9%) were grouped into high SMI, and 93 patients (42.1%) were grouped into low SMI, as shown in Table 2. In the high SMI group, male recipients accounted for a smaller proportion than in the low SMI group (*p* = 0.045), and BMI was significantly higher than that in the low SMI group (*p* = 0.011). The high SMI group had higher KPS scores (*p* < 0.001), lower serum ammonia levels (*p* = 0.044), and lower Child–Pugh scores (*p* < 0.001) and MELD scores (*p* = 0.015) than the low SMI group. A significantly decreased rate of encephalopathy and ascites was observed in patients with high SMI (*p* < 0.001 and *p* < 0.001). The other characteristics were comparable between the two populations.

The high SMI group had an overwhelming advantage in short-term outcomes, such as a lower incidence of postoperative infection (*p* = 0.001), a lower 90 day CCI (*p* < 0.001), and a shorter duration of ICU stay (*p* = 0.003). Meanwhile, the high SMI group had a higher OS (1, 3, and 5 year OS: 97.7%, 87.5%, and 79.4% vs. 92.4%, 77%, and 55.4%, *p* = 0.001, Figure 3c) and RFS (1, 3, and 5 year RFS: 95.3%, 83.8%, and 76.2% vs. 76.5%, 64.7%, and 61.7%, *p* = 0.001, Figure 3f) than the low SMI group.

### 3.5. Outcome Analyses according to CAS Grade

After stratification by CAS grade, 46 patients (20.8%) were classified into CAS grade 1, 96 patients (43.4%) were classified into CAS grade 2, and 79 patients (35.7%) were classified into grade 3 (Table 3). Patients in CAS grade 1 were associated with the significantly lowest percentage of male recipients, encephalopathy, and ascites, the highest KPS score, and the lowest Child–Pugh score and MELD score. Moreover, there was a relatively significant relationship among patients in CAS 1, 2 and 3, with increasing CAS grade, an ascending trend toward postoperative infection, 90 day CCI, and duration of ICU stay. Moreover, after post hoc analysis, we found that the ALB level and 90 day CCI were statistically significant between every two groups. Patients with CAS grade 1 were associated with the greatest OS and RFS, whereas patients with CAS grade 3 had contrary outcomes (1, 3, and 5 year OS for CAS grades 1, 2, and 3: 97.8%, 93.4%, and 87.9% vs. 96.9%, 83.3%, and 73.5% vs. 92.3%, 76.7%, and 53.5%, *p* < 0.001, Figure 4a) and (1, 3, and 5 year RFS for CAS grades 1, 2, and 3: 95.7%, 86.9%, and 82.5% vs. 93.6%, 82.2%, and 70.8% vs. 73.7%, 59.6%, and 57.9%, *p* = 0.001, Figure 4b).

### 3.6. ROC Curve Analysis and risk Factor Analysis

Figure 5 indicates that the CAS had a more accurate identification ability for OS than AGS and SMI (AUC: 0.710 vs. 0.618 and 0.646, respectively) (Figure 5a). Similarly, the AUC of CAS grade (0.700) was greater than that of AGS (0.620) and SMI (0.612) for RFS (Figure 5b). Lastly, our univariable Cox proportional hazards regression model showed a significant relevance of male recipient, NLR > 2.6, Child–Pugh C, MELD score > 20, microvascular invasion and elevated CAS grade with OS (Table 4). The multivariable analysis identified male recipient (HR: 1.824, 95% CI: 1.349–2.502, *p* = 0.017), Child–Pugh C (HR: 2.045, 95% CI: 1.028–4.426, *p* = 0.011), MELD score > 20 (HR: 1.984, 95% CI: 1.113–3.026, *p* = 0.025), CAS grade 2 (HR: 3.045, 95% CI: 1.382–6.896, *p* = 0.001), and CAS grade 3 (HR: 4.412, 95% CI: 2.117–9.480, *p* < 0.001) as independent factors associated with impaired OS.

## 4. Discussion

This study comprehensively explored the association of malnutrition with short- and long-term post-DDLT patient survival outcomes by evaluating the CAS grade, which is a novel prognostic marker combined with AGS and SMI. Interestingly, on the basis of our data, the prognostic value of CAS was proven to be more accentuated than that of either alone. The CAS grade reflected the nutritional and inflammatory status of HCC patients simultaneously. As an independent prognostic risk factor for OS and RFS in HCC patients, CAS grade had higher accuracy in predicting OS and RFS than AGS and SMI.

Chronic inflammation is a critical contributor to tumor development, proliferation, and metastasis and is also related to the risk of death and recurrence among malignant patients after surgery [7,8,9]. Serum ALB, produced by the liver, reflects nutritional status and participates in the body’s natural defense activities. Furthermore, low serum ALB levels were also reported to be related to chronic inflammation, which is not only associated with a reduction in circulating albumin concentrations but also probably through increased degradation, especially in patients with viral hepatitis cirrhosis [35]. GLB, produced by immune organs, is a major component of systemic inflammation and comprises numerous proinflammatory proteins. High levels of GLB resulting from immunoglobulins and acute-phase protein aggregation may be associated with the malignant microenvironment [36]. Zhang et al. found that a high GLB level was significantly related to high AFP, cirrhosis, major tumor size, and poor Edmondson grade of the tumor [37].

Although ALB and GLB are important predictive factors in many malignant tumors, their serum levels are affected by many factors, such as stress response, liver insufficiency, and alteration of body fluid volume. Therefore, their clinical value for predicting cancer patient prognosis is limited. Then, the AGR, defined as ALB (g/L)/GLB (g/L), proved to be an independent prognostic factor in digestive system cancers, upper-tract urothelial carcinoma, and oral squamous cell carcinoma [7,15,38]. Consistent with our results, patients with an AGR > 1.47 had a better survival outcome than those with an AGR ≤ 1.47 (Figure 3a,d). Meanwhile, on the basis of ALB, GLB, and AGR, the AGS has been proposed as another novel model to predict the prognosis of cancer. Li et al. compared the prognostic value of AGR and AGS in a cohort study of 458 esophageal squamous cell carcinoma (ESCC) patients and concluded that AGS outperformed AGR as a prognostic factor in ESCC [18]. Later, it was also shown that AGS could reflect the OS and RFS of non-small-cell lung cancer and cholangiocarcinoma patents, and that the predictive performance was better than that of AGR [16,17]. Similarly, our study initially confirmed this perspective in HCC patients who underwent DDLT, and the AGS was significantly related to preoperative general status, serum AFP level, Child–Pugh score, and MELD score. Additionally, the AGS showed a great capacity for predicting the long-term survival outcome for HCC patients.

Malnutrition is a common comorbidity in patients with liver cirrhosis. In our study population, 81% of patients had liver cirrhosis, which makes most HCC patients have no chance of achieving anatomical resection. Sarcopenia, a complex syndrome characterized by progressive decreases in skeletal muscle mass and function, has now been integrated into the definition of malnutrition. To date, Chinese diagnostic criteria for sarcopenia based on the L3-SMI have not been established [39]. Because the etiology of HCC and the characteristics of cirrhotic patients are markedly different in China than in other districts, we set up the cutoff value of SMI in a Chinese cohort with HCC after DDLT (male: 43.1 cm^2^/m^2^, female: 32.9 cm^2^/m^2^). The cutoff values were smaller than those determined by a North American expert (male: 50 cm^2^/m^2^, female: 39 cm^2^/m^2^) [40]. Some scholars showed that sarcopenia has a negative effect on long-term prognosis following liver transplantation [26,27,28]. However, others argued that sarcopenia was not associated with impaired survival after liver transplantation [30]. Our study showed that low SMI was a poor prognostic indicator in terms of both OS and RFS. The controversy may be attributed to differences in ethnicity, selection bias of the study populations, and the inconsistent definition of sarcopenia.

Currently, cross-sectional imaging studies are the gold standard for quantitating skeletal muscle. These measurements are not influenced by the presence of ascites or edema, especially in our study populations [21]. L3-SMI, as the optimal parameter to assess sarcopenia, has been shown to be the best correlation with whole-body muscle mass [30]. Interestingly, in our study, there was no apparent relationship in BMI between low SMI and high SMI. Similarly, Judith et al. deemed that sarcopenia is not exclusively present in patients with a low BMI and may be present as an occult condition in HCC patients with any BMI [25]. Therefore, a surgeon’s decision is fraught with the subjectivity of health status, which some clinicians call “the eyeball test” [26]. Loss of muscle mass can be precipitated by a superimposed pathological condition, such as cancer or chronic diseases [41]. The pathogenesis of sarcopenia includes systemic inflammation, myostatin signaling, and insulin resistance [16,41]. In addition, several mechanisms related to cirrhosis likely contribute to muscle alterations, such as hypoalbuminemia, hepatocyte dysfunction, and hyperammonemia [41]. Muscle acts as a metabolic partner for the liver; in turn, decreased muscle mass worsens hyperammonemia. Ammonia-lowering therapies have been shown to reverse skeletal muscle alterations in hyperammonemic rodent models [42]. Therefore, lowering ammonia prior to surgery may be beneficial for a better prognosis.

To comprehensively evaluate the impact of nutritional status and inflammatory environment on the prognosis of DDLT in HCC patients, given the prognostic value of AGS and SMI, a novel index (CAS grade) was introduced. It exhibits greater correlations with OS and RFS than each alone. Male recipient, a dependent risk factor for poor survival outcome, occupied an increasing percentage with CAS grade elevation. One explanation is that there is a clear sex predisposition for sarcopenia in cirrhosis, being more prevalent in males than in female patients. Fluctuations in hormone levels lead to more and faster loss of skeletal muscle [27]. The other explanation is that males have significantly more visceral fat, whereas females have more subcutaneous fat. Subcutaneous adipose tissue is the major producer of leptin, the hormone that regulates insulin sensitivity, glucose and lipid metabolism, and the immune response [43]. KPS scoring, as an assessment of the overall performance status of patients, is significantly related to CAS grade. Despite its subjectivity, Paul et al. believed that the KPS is perhaps a reflection of the overall physical and mental status of patients with end-stage liver disease that could not be quantified by objective parameters [44]. However, the KPS score may lack reliability in this study, where the large difference in the presence of encephalopathy and ascites between different CAS grades could influence the KPS score assessment.

The MELD score and Child–Pugh score are the most widely used for evaluating donor allocation and liver function, respectively. Specifically, they play important roles in predicting prognosis for HCC patients and are dependent risk factors for poor outcome. Despite the irrefutable benefits of the MELD score, the limitations of MELD score have been recognized, and there are ongoing attempts to improve it [21,30]. One of the major limitations of the MELD score is the lack of evaluation of the nutritional and functional status of patients on the waiting list. Furthermore, the present data suggest that the relationship between low muscle mass and poor outcome is independent of the MELD score [15,41]. This result is consistent with our findings. Tandon et al. showed that sarcopenic patients with a low MELD score had a similar outcome compared with patients with a high MELD score with or without sarcopenia [45]. In addition, a Japanese study also included measures of skeletal muscle in the MELD score (Muscle-MELD score) to predict mortality after living donor liver transplantation (LDLT) [46]. Therefore, enrolling the CAS grade in the MELD score may be used to more accurately select patients in the waiting list and allocate organs in the future. In other words, HCC patients who conformed to the UCSF criteria concurrent with CAS grade 0 had priority to receive the graft in terms of utilization value. The identification of patients listed for LT who are susceptible to increased postoperative morbidity and mortality is pivotal. We comprehensively analyzed the impact of CAS on postoperative complications by 90-day CCI, finding that CAS grade 3 is significantly associated with poorer short-term outcomes, especially in the occurrence of infection episodes. Infectious complications are significant sources of mortality for liver transplant recipients. Krell et al. claimed that increased vulnerability to infection was associated with sarcopenia, but the potential influence of sarcopenia on infection-related outcomes deserves further investigation [29].

Certainly, this work must be considered within the context of its limitations. Firstly, we reported on a retrospective study with a cohort of patients from a single center. Future prospective studies should include a wider ethnicity and multiple institutions so as to set the optimized cutoffs for male and female in the larger population. Secondly, selection bias for patient inclusion was present in the study group. Patients with allograft nonfunction and no discernible cause leading to retransplantation or death were excluded. Thirdly, ALB and GLB were the latest laboratory tests prior to the surgical procedure, and they might not reflect the actual situation due to albumin infusion. Fourthly, we need to consider whether our cutoff values for ALB, GLB, and SMI were adequate to define the CAS grade in a slightly insufficient population and the deficiency for using alone CAS as a prognostic parameter in HCC patients who underwent DDLT. Lastly, we had no comprehensive data about the patients’ mobility after surgery; hence, we could not objectively evaluate their self-care ability. Until now, there has been no multicenter prospective study of a Chinese cohort to provide a definition of CAS grade.

Notwithstanding the aforementioned limitations. To the best of our knowledge, this was the first study to integrate preoperative ALB, GLB, and skeletal muscle mass to predict short- and long-term outcomes of HCC patients who underwent DDLT. Assessing CAS grade in possible LT candidates can help to predict posttransplant outcomes. Therefore, CAS grade can be supplemented in the process of recipient selection and organ allocation.

## 5. Conclusions

The present study provided a novel prognostic index combining preoperative AGS and SMI that was closely related to postoperative short-term and long-term outcomes for HCC patients who underwent DDLT. Performing CAS grade evaluation may be used for clinical decision making. Meanwhile, it is necessary for nutritionists to perform nutritional status assessment and nutrition support therapy before liver transplantation. 

## Figures and Tables

**Figure 1 jcm-12-02237-f001:**
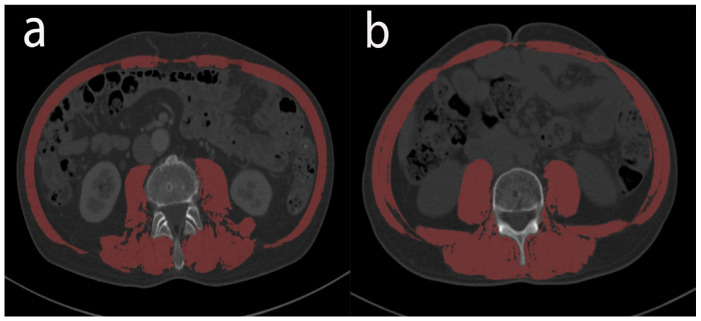
Cross-sectional computed tomography images of the third lumbar vertebra used to measure areas of total skeletal muscle (red shadows): low SMI (**a**); high SMI (**b**). SMI, skeletal muscle index.

**Figure 2 jcm-12-02237-f002:**
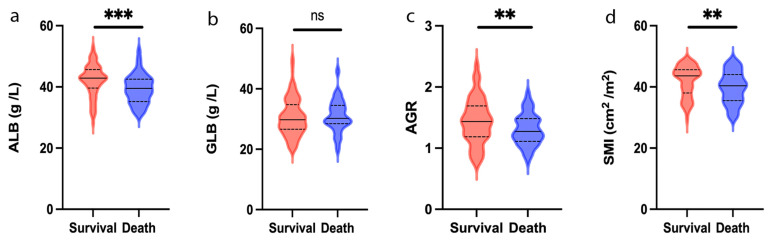
Violin plots showing the preoperative ALB (**a**), GLB (**b**), AGR (**c**), and SMI (**d**) level in survival and death group at the end of follow-up. Solid lines represent the median value; dotted lines represent quartiles. ** *p* < 0.01; *** *p* < 0.001; ns, no significance; ALB, albumin; GLB, globulin; AGR, albumin-to-globulin ratio; SMI, skeletal muscle index.

**Figure 3 jcm-12-02237-f003:**
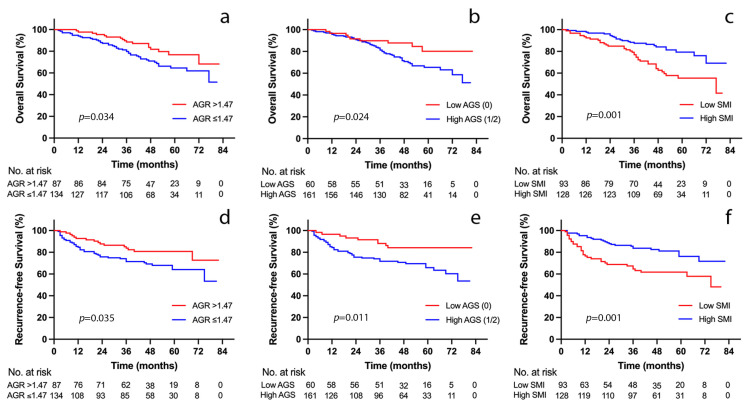
Kaplan–Meier curves for overall survival and recurrence-free survival stratified by AGR (**a**,**d**), AGS (**b**,**e**), and SMI (**c**,**f**) in the study cohort. AGR, albumin-to-globulin ratio; AGS, albumin–globulin score; SMI, skeletal muscle index.

**Figure 4 jcm-12-02237-f004:**
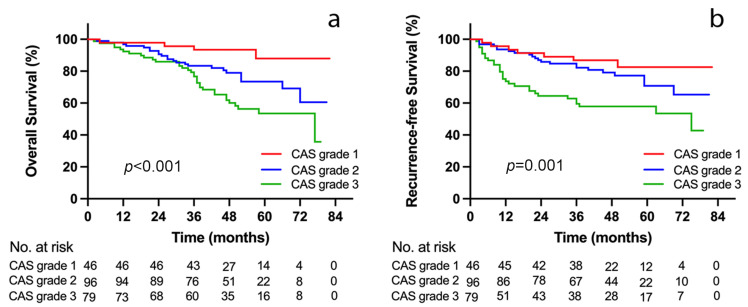
Kaplan–Meier curves for overall survival (**a**) and recurrence-free survival (**b**) stratified by CAS grade in the study cohort. CAS, combination of albumin–globulin score and skeletal muscle index.

**Figure 5 jcm-12-02237-f005:**
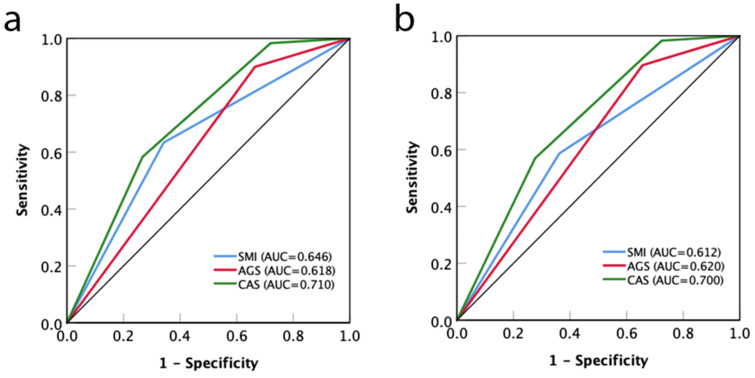
Comparison of AUC for AGS, SMI, and CAS grade in predicting overall survival (**a**) and recurrence-free survival (**b**) in the study cohort. AUC, area under curve; AGS, albumin–globulin score; SMI, skeletal muscle index; CAS, combination of albumin–globulin score and skeletal muscle index.

**Table 1 jcm-12-02237-t001:** Characteristics of included HCC patients who underwent DDLT for male and female.

Variables	Male (*n* = 187)	Female (*n* = 34)
Demographic and anthropometric characteristics in recipients		
Age, years, median (range)	50 (18–69)	50 (21–69)
BMI, kg/m^2^, median (range)	22.7 (13.9–33.6)	22.4 (15.2–30.9)
SMI, cm^2^/m^2^, median (range)	43.7 (32–49.3)	35.6 (28.6–41.3)
KPS, %, median (range)	80 (10–100)	70 (10–90)
HBV, *n* (%)	162 (86.6)	26 (76.5)
HCV, *n* (%)	9 (4.8)	4 (11.8)
Alcohol, *n* (%)	11 (5.9)	1 (2.9)
Nonalcoholic steatohepatitis, *n* (%)	2 (1.1)	2 (5.9)
Diabetes mellitus, *n* (%)	34 (18.2)	4 (11.8)
Encephalopathy, *n* (%)	30 (16.0)	5 (14.7)
Ascites, *n* (%)	51 (27.3)	7 (20.6)
Demographic and anthropometric characteristics in donors		
Age, years, median (range)	43 (23–64)	40 (24–60)
Male, *n* (%)	114 (61.0)	9 (26.5)
BMI, kg/m^2^, median (range)	23.0 (16.6–31.5)	21.9 (16.4–25.9)
DRI, median (range)	1.3 (0.8–1.9)	1.4 (0.7–1.9)
Laboratory parameters		
ALT, IU/L, median (range)	38 (6–298)	31 (7–180)
AST, IU/L, median (range)	41 (11–327)	42.5 (13–524)
Platelet count, ×10^9^/L, median (range)	74 (23–418)	71 (14–501)
Ammonia, μmol/L, median (range)	58 (16–327)	53.5 (24–169)
Total bilirubin, μmol/L, median (range)	20.5 (8.4–731.7)	24.9 (10.9–468.9)
INR, median (range)	1.16 (0.83–2.07)	1.19 (0.93–3.45)
Creatinine, μmol/L, median (range)	71 (30–294)	53 (36–216)
ALB, g/L, median (range)	42.2 (28.6–53.2)	42.0 (27.9–50)
GLB, g/L, median (range)	29.8 (19.6–50.4)	29.3 (20.5–42.3)
AGR, median (range)	1.41 (0.73–2.37)	1.43 (0.76–2.16)
AGS, *n* (%)		
Low (0)	49 (26.2)	11 (32.4)
High (1/2)	138 (73.8)	23 (67.6)
NLR, median (range)	3.06 (0.37–24.81)	2.90 (0.71–9.99)
Child–Pugh score, median (range)	6 (5–12)	6 (5–12)
Child–Pugh A/B/C, *n* (%)	119 (63.6)/52 (27.8)/16 (8.6)	22 (64.7)/8 (23.5)/4 (11.8)
MELD score, median (range)	7 (2–33)	6 (2–27)
Serum AFP ≥ 400 ng/mL, *n* (%)	34 (18.2)	9 (26.5)
Intraoperative parameters		
Cold ischaemic time, min, median (range)	500 (405–755)	505 (410–720)
Warm ischaemic time, min, median (range)	50 (42–60)	47 (40–57)
Red blood cell transfusions, unit, median (range)	9 (0–23)	7 (0–17)
Fresh frozen plasma transfusions, mL, median (range)	1000 (0–2850)	875 (0–3100)
Histological and gross features of tumors		
Solitary tumor, *n* (%)	116 (62.2)	25 (73.5)
Largest tumor size, cm, median (range)	3.2 (0.5–6.5)	4 (1–6.5)
Total tumor size, cm, median (range)	4 (0.5–8)	4.75 (1–8)
Fibrosis, *n* (%)		
Early (Ishak 1–2)	6 (3.2)	3 (8.8)
Intermediate (Ishak 3–4)	27 (14.4)	6 (17.6)
Advanced; cirrhosis (Ishak 5–6)	154 (82.4)	25 (73.5)
Differentiation of HCC, *n* (%)		
Well	10 (5.3)	3 (8.8)
Moderate	119 (63.6)	21 (61.8)
Poor	58 (31.0)	10 (29.4)
Microvascular invasion, *n* (%)	61 (32.6)	9 (26.5)
Prognostic outcome		
Postoperative infection, *n* (%)	56 (29.9)	9 (26.5)
90 day CD ≥ 3 complications, *n* (%)	82 (44)	14 (41)
90 day CCI, median (range)	46.2 (8.7–100)	44.3 (8.7–88.6)
90 day mortality, *n* (%)	2 (1.1)	0 (0)
ICU stay, d, median (range)	5 (1–65)	4 (2–21)
Postoperative hospital stay, days, median (range)	16 (8–98)	16.5 (9–39)

DDLT deceased donor liver transplantation, HCC hepatocellular carcinoma, BMI body mass index, SMI skeletal muscle index, KPS Karnofsky performance score, HBV hepatitis B virus, HCV hepatitis C virus, DRI donor risk index, ALT alanine aminotransferase, AST aspartate aminotransferase, INR international normalized ratio, ALB albumin, GLB globulin, AGR albumin-to-globulin ratio, AGS albumin–globulin score, NLR neutrophil-to-lymphocyte ratio, MELD model for end-stage liver disease, AFP alpha-fetoprotein, CD Clavien–Dindo classification, CCI comprehensive complication index, ICU intensive care unit.

**Table 2 jcm-12-02237-t002:** Correlation among AGS, SMI, and clinicopathological characteristics of HCC patients who underwent DDLT.

Variables	AGS			SMI		
	Low (0)	High (1/2)	*p*-Value	Low	High	*p*-Value
Total patients	60	161	--	93	128	--
Recipient age, years, median (range)	50.5 (18–69)	50 (21–69)	0.368	50 (21–68)	49 (18–69)	0.737
Recipient gender, male, *n* (%)	49 (81.7)	138 (85.7)	0.458	84 (90.3)	103 (80.5)	0.045
Recipient BMI, kg/m^2^, median (range)	22.6 (16.5–29.8)	22.7 (13.9–33.6)	0.921	21.8 (13.9–27.7)	23.1 (15.2–33.6)	0.071
KPS, %, median (range)	80 (10–100)	70 (10–100)	0.039	70 (10–100)	80 (20–100)	<0.001
Diabetes mellitus, *n* (%)	8 (13.3)	30 (18.6)	0.353	16 (17.2)	22 (17.2)	0.997
Encephalopathy, *n* (%)	4 (6.7)	31 (19.3)	0.023	25 (26.9)	10 (7.8)	<0.001
Ascites, *n* (%)	10 (16.7)	48 (29.8)	0.048	38 (40.9)	20 (15.6)	<0.001
Ammonia, μmol/L, median (range)	51 (19–218)	62 (16–327)	0.112	69 (18–327)	53 (16–218)	0.044
ALB, g/L, median (range)	44.3 (36–53.2)	40.6 (27.9–52)	<0.001	39.6 (27.9–52)	43.2 (30.9–53.2)	<0.001
GLB, g/L, median (range)	25.9 (19.7–37.5)	31.7 (19.6–50.4)	<0.001	30.9 (19.6–50.4)	29.4 (19.7–48.5)	0.087
NLR, median (range)	2.88 (0.53–24.81)	3.23 (0.37–17.38)	0.354	3.28 (0.65–24.81)	2.95 (0.37–16.65)	0.053
Child–Pugh score, median (range)	5 (5–11)	6 (5–12)	0.002	7 (5–12)	5 (5–12)	<0.001
MELD score, median (range)	7 (2–27)	8 (2–33)	0.037	8 (2–33)	6 (2–25)	0.015
Serum AFP ≥ 400 ng/mL, *n* (%)	5 (8.3)	38 (23.6)	0.011	19 (20.4)	24 (18.8)	0.755
Multiple tumor, *n* (%)	17 (28.3)	63 (39.1)	0.137	31 (33.3)	49 (38.3)	0.450
Total tumor size, cm, median (range)	4 (0.5–8)	4.3 (1–8)	0.152	4.7 (0.5–8)	4 (1–8)	0.062
Liver cirrhosis, *n* (%)	49 (81.7)	130 (80.7)	0.525	78 (83.9)	101 (78.9)	0.353
Differentiation of HCC, *n* (%)						
Well	5 (8.3)	8 (5.0)	0.344	6 (6.5)	7 (5.5)	0.759
Moderate	42 (70)	98 (60.9)	0.210	53 (57.0)	87 (68.0)	0.094
Poor	13 (21.7)	55 (34.2)	0.073	34 (36.6)	34 (26.6)	0.112
Microvascular invasion, *n* (%)	16 (26.7)	54 (33.5)	0.329	32 (34.4)	38 (29.7)	0.456
Postoperative infection, *n* (%)	10 (16.7)	55 (34.2)	0.011	38 (40.9)	27 (21.1)	0.001
90 day CCI, median (range)	33.7 (8.7–88.6)	56.1 (8.7–100)	<0.001	59.9 (26.2–100)	42.4 (8.7–100)	<0.001
ICU stay, days, median (range)	4 (1–43)	6 (1–65)	0.039	7 (1–65)	4 (1–24)	0.003

AGS albumin–globulin score, SMI skeletal muscle index, DDLT deceased donor liver transplantation, HCC hepatocellular carcinoma, BMI body mass index, KPS Karnofsky performance score, ALB albumin, GLB globulin, NLR neutrophil-to-lymphocyte ratio, MELD model for end-stage liver disease, AFP alpha-fetoprotein, CCI comprehensive complication index, ICU intensive care unit.

**Table 3 jcm-12-02237-t003:** Correlation between CAS and clinicopathological characteristics in HCC patients who underwent DDLT.

Variables	CAS			
	Grade 1	Grade 2	Grade 3	*p*-Value
Total patients	46	96	79	--
Recipient age, years, median (range)	52.5 (28–63)	50 (21–69)	49 (18–69)	0.368
Recipient gender, male, *n* (%)	35 (76.1)	79 (82.3)	73 (92.4)	0.036
Recipient BMI, kg/m^2^,	23.6 (19.8–29.8)	22.7 (16.9–33.6)	21.3 (13.9–29.4)	0.521
KPS, %, median (range)	80 (50–100)	80 (10–100)	70 (10–100)	0.039
Diabetes mellitus, *n* (%)	6 (13.0)	17 (17.7)	15 (19.0)	0.686
Encephalopathy, *n* (%)	3 (6.5)	11 (11.5)	21 (26.6)	0.004
Ascites, *n* (%)	4 (8.7)	22 (22.9)	32 (40.5)	<0.001
Ammonia, μmol/L, median (range)	45 (19–218)	58 (16–171)	67.5 (18–327)	0.092
ALB, g/L, median (range)	46.4 (40.7–53.2)	42.1 (30.9–51.2)	38.3 (27.9–52)	<0.001
GLB, g/L, median (range)	26.2 (19.7–31.3)	30.7 (20.5–48.5)	32.7 (19.6–50.4)	<0.001
NLR, median (range)	2.62 (0.53–15.65)	3.21 (0.37–24.81)	3.24 (0.65–17.38)	0.097
Child-Pugh score, median (range)	5 (5–10)	6 (5–12)	7 (5–12)	0.002
MELD score, median (range)	7 (2–25)	8 (2–24)	8 (2–33)	0.037
Serum AFP ≥ 400 ng/mL, *n* (%)	5 (10.9)	19 (19.8)	19 (24.1)	0.198
Multiple tumor, *n* (%)	14 (30.4)	39 (40.6)	27 (34.2)	0.446
Total tumor size, cm, median (range)	4 (1–8)	4 (0.5–8)	4.8 (1–8)	0.152
Liver cirrhosis, *n* (%)	38 (82.6)	73 (76.0)	68 (86.1)	0.231
Differentiation of HCC, *n* (%)				
Well	4 (8.7)	4 (4.2)	5 (6.3)	0.550
Moderate	32 (69.6)	64 (66.7)	44 (55.7)	0.201
Poor	10 (21.7)	28 (29.2)	30 (38.0)	0.149
Microvascular invasion, *n* (%)	11 (23.9)	27 (28.1)	32 (40.5)	0.096
Postoperative infection, *n* (%)	6 (13.0)	27 (28.1)	32 (40.5)	0.005
90-day CCI, median (range)	33.7 (8.7–68.6)	46.2 (26.2–100)	59.9 (26.2–100)	<0.001
ICU stay, d, median (range)	4 (1–21)	4.5 (1–43)	8 (1–65)	0.039

CAS combination of albumin-globulin score and skeletal muscle index, DDLT deceased donor liver transplantation, HCC hepatocellular carcinoma, BMI body mass index, KPS Karnofsky performance score, ALB albumin, GLB globulin, NLR neutrophil-to-lymphocyte ratio, MELD model for end-stages liver disease, AFP Alpha-fetoprotein, CCI comprehensive complication index, ICU intensive care unit.

**Table 4 jcm-12-02237-t004:** Univariate and multivariate analysis identify independent prognostic factors for overall survival in the cohort.

Variables	Univariate Analysis		
	HR	95% CI	*p*-Value
Recipient Age (>60 years)	1.229	0.883–1.886	0.335
Recipient gender (male)	2.196	1.173–4.394	0.015
KPS (C)	1.214	0.864–2.045	0.371
Encephalopathy	1.230	0.865–1.990	0.407
Ascites	1.340	0.505–2.142	0.173
NLR (>2.6)	1.873	1.384–3.014	0.037
Child–Pugh C	2.017	1.538–3.873	0.016
MELD score (>20)	1.776	0.984–2.659	0.086
Serum AFP (>400 ng/mL)	1.234	0.488–2.790	0.336
Multiple tumors	1.432	0.871–2.232	0.116
Meeting Milan criteria	0.730	0.559–1.866	0.245
Liver cirrhosis	1.098	0.700–1.959	0.572
Differentiation of HCC (poor)	1.398	0.514–2.359	0.135
Microvascular invasion	1.710	0.877–2.346	0.095
CAS grade (2)	3.391	2.028–7.135	<0.001
CAS grade (3)	4.031	2.123–7.574	<0.001
	Multivariate analysis		
	HR	95% CI	*p*-value
Recipient gender (male)	1.824	1.349–2.502	0.017
NLR (>2.6)	1.485	0.892–2.449	0.087
Child–Pugh C	2.045	1.028–4.426	0.011
MELD score (>20)	1.984	1.113–3.026	0.025
Microvascular invasion	1.290	0.884–1.857	0.120
CAS grade (2)	3.045	1.382–6.896	0.001
CAS grade (3)	4.412	2.117–9.480	<0.001

HCC hepatocellular carcinoma, HR hazard risk, CI confidence interval, KPS Karnofsky performance score, NLR neutrophil-to-lymphocyte ratio, MELD model for end-stage liver disease, AFP alpha-fetoprotein, CAS combination of albumin–globulin score and skeletal muscle index.

## Data Availability

Data are contained within the article and are available from the corresponding author on reasonable request.

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
