# Peer review of "Albumin–Globulin Score Combined with Skeletal Muscle Index as a Novel Prognostic Marker for Hepatocellular Carcinoma Patients Undergoing Liver Transplantation"

_jcm, 2023, doi:10.3390/jcm12062237_

Round 1

Reviewer 1 Report

Major opinion

    The major opinion is about the Table 1 and Figure 2. In my opinion, the table one could be categorized into survival and death (this means figure 2 is integrated into a table). The comparison analysis can be conducted in this table and P value can be showed. This also emphasizes the albumin (ALB), albumin-globulin ratio (AGR), and skeletal muscle index (SMI) differences between survival and death groups. Than receiver operating characteristic (ROC) curve figures can be expressed. Finally, the optimal cutoff values are generated by Youden’s test.

    If I do not misunderstand, the targeting status of ROC curve is “2-year survival status” (Line 180-181). Therefore, the first issue in this article is to show the important role of ALB, AGR, and SMI in survival. I have no idea why authors categorizing by sex in Table 1. I also notice the table by sex groups in reference 40 (Hepatology. 2019;70(5):1816-29). Authors may imitate this expression but I think the study aims different. The optimal cutoff value of SMI that must be defined separately between male and female for sarcopenia is proved by previous studies (such as reference 40). The aim of present study is to prove the roles of albumin, globulin, and sarcopenia in patients with deceased donor liver transplantation and hepatocellular carcinoma (HCC).

Minor opinion:

1.      In section Introduction, authors describe the HCC and cirrhosis nature. However, the Reference 2 [Hepatology. 2022;75(6):1402-19.] is an intrahepatic cholangiocarcinoma study. This citation is not feasible.

2.      The citation form of reference 26 is incorrect.

3.      In section Methods (Line 105), authors mentioned a factor “etiology of disease”. The “disease” is not precise. I suggest using a specific medical issue (e.g. cirrhosis, hepatitis).

4.      In section Methods (Line 111), author declare that all patients received head, chest, and abdominal computed tomography (CT) one week before liver transplantation. All organ sources were from deceased donor. This means every organ was processed via brain death evaluation, donor-recipient matching, and organ allocation. In my personal experience. It is nearly impossible to perform a delicate timing of advanced image (one week before surgery) in such patients. Did these patient really take image just one week before liver transplantation or there is a time interval (e.g. at lease within 3 months before surgery while every patient undergoing regular image follow-up every 3 months)?

5.      In section Methods (Line 125), author use “donor risk index” for a factor. I suggest that author must cite a reference for this index.

6.      In section Methods (Line 148), one factor that CT examined is “duct anatomy”. Does this mean hepatic/biliary duct? Or other specific anatomy?

7.      In section Methods (Line 165-166), the follow-up includes tumor marker and immunosuppressant blood concentration. I suggest the specific item should be described (e.g. alpha-fetoprotein, FK506, etc.).

8.      In section Methods (Line 175-176), both parametric and non-parametric methods are conducted. First, while a parametric method was introduced, the test of normality should also be conducted for raw data (e.g. Shapiro-Wilk test etc.). Second, I notice that all data are expressed by median with range. Does this mean all comparison analysis are conducted by Mann-Whitney U method? Or authors had another consideration?

9.      In section Result (Line 197), I think that the “male” is “female”.

10.  In section Result (Line 216), What is “ALR”?

11.  In Table 3, three groups are compared together by certain statistical methods. First, authors must declare the method for analysis in section Methods. Second, post-hoc should also be conducted to analyze the data between every two groups.

Author Response

Thank you for your valuable suggestions and pertinent comments. The following are the responses and revisions I have made in response to the questions and suggestions on an item-by-item basis.

Reviewer 1:

Major opinion

    The major opinion is about the Table 1 and Figure 2. In my opinion, the table one could be categorized into survival and death (this means figure 2 is integrated into a table). The comparison analysis can be conducted in this table and P value can be showed. This also emphasizes the albumin (ALB), albumin-globulin ratio (AGR), and skeletal muscle index (SMI) differences between survival and death groups. Than receiver operating characteristic (ROC) curve figures can be expressed. Finally, the optimal cutoff values are generated by Youden’s test.

If I do not misunderstand, the targeting status of ROC curve is “2-year survival status” (Line 180-181). Therefore, the first issue in this article is to show the important role of ALB, AGR, and SMI in survival. I have no idea why authors categorizing by sex in Table 1. I also notice the table by sex groups in reference 40 (Hepatology. 2019;70(5):1816-29). Authors may imitate this expression but I think the study aims different. The optimal cutoff value of SMI that must be defined separately between male and female for sarcopenia is proved by previous studies (such as reference 40). The aim of present study is to prove the roles of albumin, globulin, and sarcopenia in patients with deceased donor liver transplantation and hepatocellular carcinoma (HCC).

We are appreciated to receive your comments. As for which grouping is suitable for Table 1, we want to clarify our reviews. At first, Table 1 is usually used to display baseline data, so as to roughly know the general situation of the research population. We also think that survival state is reasonable grouping standard, but considering the progressive writing style of manuscript, we think it is a bit inappropriate to use the main outcome as the classification point at the beginning of the manuscript. Besides, if Table 1 is grouped by survival status, it will be partially repeated with the following univariate analysis results. Second, because SMI indicator varied greatly between male and female, and we included the donor's demographic and anthropometric characteristics in Table 1. Therefore, we chose the gender as the grouping basis. Finally, as reviewer said, the aim of present study is to prove the roles of albumin, globulin, and sarcopenia in HCC patients with deceased donor liver transplantation. We think it is justifiable to take Figure 2 alone to show the importance of ALB, GLB, AGR and SMI and pave the way for CAS, a new prognostic indicator. Therefore, we hope to have more discussions with reviewer on this issue.

Minor opinion:

  1. In section Introduction, authors describe the HCC and cirrhosis nature. However, the Reference 2 [Hepatology. 2022;75(6):1402-19.] is an intrahepatic cholangiocarcinoma study. This citation is not feasible.

Thank you for pointing out our mistake. We have canceled the reference (line 39).

  1. The citation form of reference 26 is incorrect.

This is really our negligence. We have revised the format of the reference (line 576).

  1. In section Methods (Line 105), authors mentioned a factor “etiology of disease”. The “disease” is not precise. I suggest using a specific medical issue (e.g. cirrhosis, hepatitis).

Thank you for your valuable suggestion. We replaced with “cirrhosis” (line 110).

  1. In section Methods (Line 111), author declare that all patients received head, chest, and abdominal computed tomography (CT) one week before liver transplantation. All organ sources were from deceased donor. This means every organ was processed via brain death evaluation, donor-recipient matching, and organ allocation. In my personal experience. It is nearly impossible to perform a delicate timing of advanced image (one week before surgery) in such patients. Did these patient really take image just one week before liver transplantation or there is a time interval (e.g. at lease within 3 months before surgery while every patient undergoing regular image follow-up every 3 months)?

In our center, all patients waiting for DDLT are admitted to the hospital through the green channel, and then the CT examination is routinely completed in the emergency department. Except for the patients who have completed the abdominal CT examination within one week during the follow-up, but the head and chest CT examination should still be completed.

  1. In section Methods (Line 125), author use “donor risk index” for a factor. I suggest that author must cite a reference for this index.

Thank you for your valuable suggestions. We have added a reference (reference 32: Feng S, Goodrich NP, Bragg-Gresham JL, Dykstra DM, Punch JD, DebRoy MA, Greenstein SM, Merion RM. Characteristics associated with liver graft failure: the concept of a donor risk index. Am J Transplant. 2006 Apr;6(4):783-90. doi: 10.1111/j.1600-6143.2006.01242.x.) in our manuscript (line 132).

  1. In section Methods (Line 148), one factor that CT examined is “duct anatomy”. Does this mean hepatic/biliary duct? Or other specific anatomy?

We are sorry that we may have not expressed it clearly. The “duct anatomy” means hepatic blood vessels and biliary ducts. We have made it clear in line 164.

  1. In section Methods (Line 165-166), the follow-up includes tumor marker and immunosuppressant blood concentration. I suggest the specific item should be described (e.g. alpha-fetoprotein, FK506, etc.)

We have revised it as “tumor markers (alpha-fetoprotein and abnormal prothrombin), blood concentration of tacrolimus (FK506)” (line 182-183).

  1. In section Methods (Line 175-176), both parametric and non-parametric methods are conducted. First, while a parametric method was introduced, the test of normality should also be conducted for raw data (e.g. Shapiro-Wilk test etc.). Second, I notice that all data are expressed by median with range. Does this mean all comparison analysis are conducted by Mann-Whitney U method? Or authors had another consideration?

As mentioned by reviewer, the test of normality is conducted for all raw data. They didn’t conform to normal distribution. Therefore, all comparison analysis were conducted by Mann-Whitney U test.

  1. In section Result (Line 197), I think that the “male” is “female”.

Thank you for pointing out our mistake, we have revised it as “female” in line 219.

  1. In section Result (Line 216), What is “ALR”?

Thank you for pointing out our mistake, the “ALR” should be corrected as “AGR” (line 241).

  1. In Table 3, three groups are compared together by certain statistical methods. First, authors must declare the method for analysis in section Methods. Second, post-hoc should also be conducted to analyze the data between every two groups.

Thank you for your valuable suggestions. First, we performed normality test and Levene’s test, then we performed Kruskal-Wallis H test and chi‐squared test for continuous variables and categorical variables respectively for comparison of three groups, followed by the SNK-q test and Bonferroni multiple comparisons test. We have added to the statistical analysis section of Methods (line 195-197). According to the reviewer's comments, we completed post-hoc analysis, and found that the ALB level and 90-day CCI are statistically significant between every two groups. We have added the results to the Results in line 311-312.

Reviewer 2 Report

The proposed study aimed to find a novel prognostic marker that affects the prognosis of HCC patients after deceased donor liver transplantation (DDLT).  They investigated the association of malnutrition with short- and long-term post-DDLT patient survival outcomes by evaluating the CAS grade, a novel prognostic marker combined with AGS and SMI. The article is well written and presented except for minor spelling mistakes. Taking statistical values according to the p-value is also positive.

Only in the discussion part, where the limitation of retrospective research is discussed, the mobility after surgery of the patients has to be integrated. This is not precisely known how the patients feel if they have helpers at home and how they do their work.

Author Response

The proposed study aimed to find a novel prognostic marker that affects the prognosis of HCC patients after deceased donor liver transplantation (DDLT).  They investigated the association of malnutrition with short- and long-term post-DDLT patient survival outcomes by evaluating the CAS grade, a novel prognostic marker combined with AGS and SMI. The article is well written and presented except for minor spelling mistakes. Taking statistical values according to the p-value is also positive. 

Only in the discussion part, where the limitation of retrospective research is discussed, the mobility after surgery of the patients has to be integrated. This is not precisely known how the patients feel if they have helpers at home and how they do their work.

Thank you for your affirmations very much. As you said, patients’ mobility after LT is a crucial issue, but our retrospective research didn’t include the relevant information completely. Therefore, we discuss it as a limitation in our manuscript (line 483-485). In addition, we found some spelling mistakes and modified them.

Reviewer 3 Report

The manuscript entitled “Albumin-globulin score combined with skeletal muscle index as a novel prognostic marker for hepatocellular carcinoma patients undergoing liver transplantation” addresses major issues in the clinical management of hepatocellular carcinoma (HCC) by the deceased donor liver transplantation (DDLT). The authors postulate CAS score, which combines albumin/globulin score (AGS) and skeletal muscle index (SMI), as a novel potential prognostic parameter for HCC after DDLT. They suggest the potential application of CAS in clinical decision-making and its potential relevance as an improving factor for the current methods used for prioritizing patients for DDLT.

The manuscript is well structured, the applied methodology is adequate for this type of study and the results are clearly presented in an informative manner. There are merely some minor issues that need to be addressed and some corrections and clarifications are required:

- Please change the phrase “in our country” with the specified country to which it refers, so that the readers do not need to check the affiliations while analyzing the article (lines 46 and 51 on page 2).

- Please elaborate why other indicators of nutritional status, which were previously reported as potential prognostic biomarkers of HCC after liver transplantation, were not analyzed in the current study (such as visceral adipose tissue and subcutaneous adipose tissue indicators, as reported in 10.1002/hep.29578 and 10.3390/jcm8101672). Also, only NLR was presented in Tables, even though LMR and PLR are reported as indicators of inflammation potentially relevant for HCC prognosis in the Introduction section (line 57).

- In line 101 on page 3, the authors state that the Informed consent was waived, which is inconsistent with the Informed Consent Statement on page 14.

- Please provide definition for Milan criteria in section 2.2.

- The cutoff values for ALB, GLB and AGR should only be stated in the Results section, not in Methods (lines 141 and 142, page 3). Hypoalbuminemia and elevated GLB levels should, therefore, be defined as “(≤ ALB cutoff value)” and (˃ GLB cutoff value)”, respectively (lines 143 and 144).

- It should be stated in section 2.4. that separate cutoff values for SMI were used for males and females.

- It is unclear why 2-year survival endpoint, instead of 5-year survival, was used for constructing ROC curves which were used for calculating cutoff values.

- “males” should be corrected to “females”, line 197, page 5.

- It is stated that preoperative ABL, GLB and SMI were used for CAS, but the titles of Tables are confusing (“... characteristics of HCC patients after liver transplantation”) and need to be rephrased, so that it becomes clear that the presented results are characteristics of HCC patients who underwent DDLT, not the postoperative biochemical and other characteristics of study participants.  

- In Table 1, “Demographic characteristics” should be changed to “Demographic and anthropometric characteristics”.

- In line 222, page 7, “(AGR ≤ 1.47)” should be changed to “with AGR ≤ 1.47”.

- In line 263, page 9, “AGS” should be changed to “SMI”.

- Even though the impact of potential ethnic differences and the adequacy of cutoff values are mentioned as limitations in the Discussion section, further elaboration would be beneficial, highlighting the limitation for universal usage of CAS as a prognostic parameter in HCC with DDLT and the necessity of setting the optimized cutoffs for both genders in large population-based studies.  

Author Response

The manuscript entitled “Albumin-globulin score combined with skeletal muscle index as a novel prognostic marker for hepatocellular carcinoma patients undergoing liver transplantation” addresses major issues in the clinical management of hepatocellular carcinoma (HCC) by the deceased donor liver transplantation (DDLT). The authors postulate CAS score, which combines albumin/globulin score (AGS) and skeletal muscle index (SMI), as a novel potential prognostic parameter for HCC after DDLT. They suggest the potential application of CAS in clinical decision-making and its potential relevance as an improving factor for the current methods used for prioritizing patients for DDLT.

The manuscript is well structured, the applied methodology is adequate for this type of study and the results are clearly presented in an informative manner. There are merely some minor issues that need to be addressed and some corrections and clarifications are required:

- Please change the phrase “in our country” with the specified country to which it refers, so that the readers do not need to check the affiliations while analyzing the article (lines 46 and 51 on page 2).

Thank you for your valuable suggestion. We replaced with “China” (line 48 and 53).

- Please elaborate why other indicators of nutritional status, which were previously reported as potential prognostic biomarkers of HCC after liver transplantation, were not analyzed in the current study (such as visceral adipose tissue and subcutaneous adipose tissue indicators, as reported in 10.1002/hep.29578 and 10.3390/jcm8101672). Also, only NLR was presented in Tables, even though LMR and PLR are reported as indicators of inflammation potentially relevant for HCC prognosis in the Introduction section (line 57).

As reviewer said, there are many indicators of nutritional status, which are potential prognostic markers of HCC patients who underwent DDLT. In our present study, we have included ALB, GLB and SMI, in our opinion, too many indicators will inevitably weaken the importance of each indicator. In addition, some studies have reported that the CAS grade was important prognostic indicator in renal cell carcinoma and intrahepatic cholangiocarcinoma (reference 16 and Mao W, Zhang N, Wang K, Hu Q, Sun S, Xu Z, Yu J, Wang C, Chen S, Xu B, Wu J, Zhang H, Chen M. Combination of Albumin-Globulin Score and Sarcopenia to Predict Prognosis in Patients With Renal Cell Carcinoma Undergoing Laparoscopic Nephrectomy). Therefore, our major purpose is to explore the relationships between CAS and HCC patients who underwent DDLT.

We described that inflammation indicators were relevant to HCC patients’ prognosis in the Introduction section. However, our present study mainly discussed the relationship between nutrition and prognosis of HCC patients after DDLT. Inflammation indicators were only used as a background introduction. The NLR is one of the most classic inflammatory prognostic factors, so we chose it as the representative of minor role.

- In line 101 on page 3, the authors state that the Informed consent was waived, which is inconsistent with the Informed Consent Statement on page 14.

Thank you for pointing out our mistake. This is really our negligence. Because in our hospital, before December 2021, retrospective research didn’t need to get approval from ethics committee, after completing the manuscript, we tried our best to notify the patient or their relatives to sign the informed consent form and supplement the ethical application. We forgotten to update the relevant information in the manuscript. We have revised it to be consistent (line 107).

- Please provide definition for Milan criteria in section 2.2.

We have provided definition for Milan criteria in line 120. (single nodule ≤ 5 cm or 2-3 nodules, each ≤ 3 cm in diameter without vascular invasion or extrahepatic metastases)

- The cutoff values for ALB, GLB and AGR should only be stated in the Results section, not in Methods (lines 141 and 142, page 3). Hypoalbuminemia and elevated GLB levels should, therefore, be defined as “(≤ ALB cutoff value)” and (> GLB cutoff value)”, respectively (lines 143 and 144).

We have deleted the cutoff values for ALB, GLB and AGR in Methods (line 147) and revised hypoalbuminemia and elevated GLB levels as “(≤ ALB cutoff value)” and (˃ GLB cutoff value)”, respectively (line 149-150).

- It should be stated in section 2.4. that separate cutoff values for SMI were used for males and females.

We have stated that that the cut-off values for SMI are used for male and female in line 171.

- It is unclear why 2-year survival endpoint, instead of 5-year survival, was used for constructing ROC curves which were used for calculating cutoff values.

Because the research period is from January 2015 to December 2019, and the last follow-up time is December 2021. Each patient had been followed up for two years, but not all for five years.

- “males” should be corrected to “females”, line 197, page 5.

Thank you for pointing out our mistake, we have revised it as “female” in line 219.

- It is stated that preoperative ABL, GLB and SMI were used for CAS, but the titles of Tables are confusing (“... characteristics of HCC patients after liver transplantation”) and need to be rephrased, so that it becomes clear that the presented results are characteristics of HCC patients who underwent DDLT, not the postoperative biochemical and other characteristics of study participants. 

We are appreciated to receive your comments. We try to modify the titles of all the Tables. The replaced title of Table 1 is “Characteristics of included HCC patients who underwent DDLT for male and female”, Table 2 is “Correlation between AGS, SMI and clinicopathological characteristics of HCC patients who underwent DDLT”, Table 3 is “Correlation between CAS and clinicopathological characteristics in HCC patients who underwent DDLT”, and Table 4 is “Univariate and multivariate analysis identify independent prognostic factors for overall survival in the cohort”.

- In Table 1, “Demographic characteristics” should be changed to “Demographic and anthropometric characteristics”.

Thank you for your valuable suggestion. We have changed “Demographic characteristics” to “Demographic and anthropometric characteristics” in Table 1.

- In line 222, page 7, “(AGR ≤ 1.47)” should be changed to “with AGR ≤ 1.47”.

We have changed “(AGR ≤ 1.47)” to “with AGR ≤ 1.47” in line 249.

- In line 263, page 9, “AGS” should be changed to “SMI”.

Thank you for pointing out our mistake. We have changed “AGS” to “SMI” in line 293.

- Even though the impact of potential ethnic differences and the adequacy of cutoff values are mentioned as limitations in the Discussion section, further elaboration would be beneficial, highlighting the limitation for universal usage of CAS as a prognostic parameter in HCC with DDLT and the necessity of setting the optimized cutoffs for both genders in large population-based studies.  

Thank you for your valuable suggestion. We have improved the relevant limitations in the Discussion section (line 475, 482-483).